# Normocephalic Children Exposed to Maternal Zika Virus Infection Do Not Have a Higher Risk of Neurodevelopmental Abnormalities around 24 Months of Age than Unexposed Children: A Controlled Study

**DOI:** 10.3390/pathogens12101219

**Published:** 2023-10-06

**Authors:** Juannicelle T. A. M. Godoi, Silvia F. B. M. Negrini, Davi C. Aragon, Paulo R. H. Rocha, Fabiana R. Amaral, Bento V. M. Negrini, Sara R. Teixeira, Aparecida Y. Yamamoto, Heloisa Bettiol, Marisa M. Mussi-Pinhata

**Affiliations:** 1Department of Pediatrics, Ribeirão Preto Medical School, University of São Paulo, Brazil, Av. Bandeirantes 3009, Campus USP, Ribeirão Preto CEP 14049-900, São Paulo, Brazil; juannitamg@gmail.com (J.T.A.M.G.); sissinegrini@gmail.com (S.F.B.M.N.); dcaragon@fmrp.usp.br (D.C.A.); paulo_higa16@hotmail.com (P.R.H.R.); biarezendeamaral@yahoo.com.br (F.R.A.); bnegrini@hcrp.usp.br (B.V.M.N.); yulie@fmrp.usp.br (A.Y.Y.); hbettiol@fmrp.usp.br (H.B.); 2Department of Imaging, Hematology and Oncology, Ribeirão Preto Medical School, University of São Paulo, Ribeirão Preto CEP 14049-900, São Paulo, Brazil; teixeiras@chop.edu; 3Department of Radiology, Children’s Hospital of Philadelphia, Philadelphia, PA 19104, USA

**Keywords:** Zika virus, pregnancy, children, neuropsychomotor development, Bayley III Screening Test

## Abstract

Although very few controlled studies are available, in utero Zika virus (ZIKV)-exposed children are considered at risk for neurodevelopmental abnormalities. We aimed to identify whether there is an excess risk of abnormalities in non-microcephalic children born to mothers with confirmed ZIKV infection compared with ZIKV-unexposed children from the same population. In a cross-sectional study nested in two larger cohorts, we compared 324 ZIKV-exposed children with 984 unexposed controls. Outcomes were assessed using the Bayley Screening Test III applied around 24 months of age. Relative risks for classifying children as emergent or at-risk for neurodevelopmental delay in at least one of five domains were calculated, adjusting for covariates. In four of the five domains, few children were classified as emergent (4–12%) or at-risk (0.3–2.16%) but for the expressive communication domain it was higher for emergent (19.1–42.9%). ZIKV-exposed children were half as frequently classified as emergent, including after adjusting for covariates [RR = 0.52 (CI 95% 0.40; 0.66)]. However, no difference was detected in the at-risk category [RR = 0.83 (CI 95% 0.48; 1.44)]. Normocephalic children exposed to the Zika virus during pregnancy do not have a higher risk of being classified as at risk for neurodevelopmental abnormalities at two years of age.

## 1. Introduction

Pooled data from population-based studies have estimated that relatively few neonates (6%;95% CI: 3–9) exposed to the Zika virus (ZIKV) infection during pregnancy have central nervous system (CNS) anomalies [1]. Unfortunately, nucleic acid and serological laboratory testing are limited in identifying congenitally infected infants [2]. Hence, all ZIKV-exposed babies can be considered at risk for potential late consequences. 

Descriptive studies have explored the neurodevelopment of ZIKV-exposed normocephalic 12–36-month-old infants using different methods. Delays in one or more developmental domains have been reported in 30% or more of the children [3,4,5,6,7,8,9,10,11,12]. However, these studies might have overestimated the risk of abnormalities related to ZIKV exposure due to the lack of a comparison group. Conflicting results have been reported comparing ZIKV-exposed and unexposed children, ranging from much poorer performance on neurodevelopmental tests [13,14,15] to mild to moderate abnormalities in a few children [16,17], or similar neurodevelopment [18,19,20]. However, most of these studies evaluated a somewhat small number of children, did not consider potential confounders, and the criteria for the absence of maternal infection was not well established. A recent meta-analysis concluded that larger controlled studies are required to confirm whether ZIKV exposure in utero is associated with adverse neurodevelopmental outcomes in children [21].

Our main objective was to determine whether normocephalic infants born to mothers with confirmed ZIKV infection had an excess risk of abnormal neurodevelopment at 24 months of age compared to their unexposed peers. We compared motor, language, and cognitive development in a cohort of ZIKV-exposed normocephalic infants with a large cohort of infants born in the same city six years before the ZIKV outbreak. 

## 2. Material and Methods

### 2.1. Study Design

This was a cohort study nested in two large cohorts of mother–infant pairs. The *ZIKV-exposed infants* were derived from the Natural History of Zika Virus Infection in Gestation (NATZIG), a prospective cohort study to characterize pregnancy and child adverse outcomes of ZIKV infection during gestation [22]. The *ZIKV-unexposed* controls consisted of children enrolled in the Brazilian Ribeirão Preto and São Luís prenatal cohort, or BRISA, to identify non-classical risk factors for preterm birth [23]. 

### 2.2. Settings and Participants

Both the NATZIG and BRISA cohorts were recruited from the Ribeirão Preto region of northeastern São Paulo State, Brazil. The NATZIG recruited ZIKV-infected pregnant women and their infants during the 2015–2016 outbreak. BRISA recruited women in prenatal services during 2010–2011, when no ZIKV was detected in Brazil. Figure 1 shows the flowchart of the study participants. 

*NATZIG cohort:* Overall, 1116 pregnant women had flavivirus-like symptoms between 2015 and 2016 and searched for care. Among these, 511 were confirmed to be infected via RNA-ZIKV testing [22] of blood and/or or urine performed at the presence at symptoms and/or amniotic fluid, fetal, or placental tissues performed at delivery. Newborns from mothers with a confirmed ZIKV diagnosis were referred for follow-up in the first 6 months of life, comprising *cohort 1 of exposed infants*. The study visits were planned at birth, 3–6 weeks, 3 ± 1, 6 ± 1, 9 ± 1, 12 ± 1, 18 ± 3, and 24 ± 3 months of age. Maternal data were obtained from the medical records. Medical history, anthropometric data, growth, neurodevelopment, feeding, immunization, and morbidity were recorded at all infant visits. Cranial ultrasound was scheduled within three months of age as previously described [24]. Hearing screening (otoacoustic emission) was performed at birth. The eyes were examined by indirect ophthalmoscopy. At each visit, the study staff instructed the mothers on how to stimulate their child’s neurodevelopment using national guidance [25]. 

Infants whose ZIKV maternal blood or urine testing performed at the presence of symptoms was delayed because of laboratory testing overload were subsequently invited for evaluation and follow-up after the maternal laboratory confirmation of the infection. They formed *cohort 2 of exposed infants* and were completely assessed at least once at around 24 months of age. 

### 2.3. Study Group Selection Criteria

We selected ZIKV-exposed non-microcephalic infants belonging to both NATZIG cohorts with an available evaluation using the Bayley Scales of Infant and Toddler Development Screening Test-3rd edition (BSIDIII Screening test or BST) [26] at around 24 months of age. Patients with other congenital infections, genetic syndromes, major malformations, birth asphyxia, birth weight (BW) < 1500 g, or major abnormalities detected on cranial ultrasound were not included in this analysis. None of the babies from the follow-up acquired health problems, such as CNS trauma or infections, that would justify exclusion from the study due to potential neurological interference. No congenital CMV screening tests were routinely performed. 

BW and head circumference (HC) measured before hospital discharge were classified according to the INTERGROWTH-21st criterion [27]. Microcephaly was defined as more than -2 z-scores from the appropriate mean for age, sex, and gestational age at birth. 

*BRISA cohort:* From February 2010 to February 2011, the Ribeirão Preto prenatal cohort recruited a convenience sample of 1400 pregnant women between 22 and 25 weeks of gestation with singleton pregnancies who had undergone a first trimester obstetric ultrasound. Trained staff obtained data on reproductive health, demographic, pregnancy characteristics, and life habits. Most children (97.8%) were re-evaluated at birth (from April 2010 to June 2011). All neonates were screened for congenital CMV infection by testing CMV DNA in saliva within three weeks of age. Mothers and children were invited for a new evaluation at the beginning of their second year of age when health conditions were reassessed. Cranial sonography was not performed on these infants.

### 2.4. Control group Selection Criteria

All toddlers with a BST performed at 13-36 months of age were initially considered eligible for inclusion in the control group. The clinical and anthropometric records were reviewed. These children were classified as having normocephalic fitting to birth [27] and 13 to 36-month-old HC measures [28]. Those identified as having health conditions that could affect their neurodevelopment were not selected (BW < 1500 g, severe neonatal asphyxia, major CNS malformation, postnatal meningitidis, epilepsy, West syndrome, cystic fibrosis, and congenital CMV infection). 

### 2.5. Ethics in Human Research

The study was conducted according to the guidelines of the Declaration of Helsinki. All NATZIG (CAAE 56522216.0.0000.5440) and BRISA (Process #11157/2008) cohorts’ study procedures received ethical approval form Ethics Committee in Human Research of Ribeirão Preto School of Medicine, University of São Paulo, Brazil. Written informed consent was obtained from all the participants.

### 2.6. Outcome Measurement

The main outcome was defined in both the study and control groups using the BST around 24 months of age. In ZIKV-exposed children, the BST was applied individually by a certified examiner from 3 to 24 months of age every 3 to 6 months. The BST was applied at a mean age of 23 (SD = 1.85) months. For the subset of all 105 ZIKV-exposed children who were being followed and seen at 12 months of age, the complete Bayley-III Scale of Infant and Toddler Development (BSIDIII) [29] was then administered. ZIKV-unexposed children were evaluated with the BST by a certified examiner within a mean of 22 (SD = 3.29) months during the cohort follow-up. No complete BSID-III test was performed in this cohort. 

### 2.7. Data Analysis

We consolidated data from the datasets of the two original cohorts. Maternal, infant, and independent variables for the primary outcome were harmonized for this analysis: age (years), complete years of schooling, alcohol or illicit drug(s) use during pregnancy (yes/no), parity (primiparous/multiparous), obstetric morbidity (hypertension and diabetes mellitus), infant sex, BW (grams), HC (cm), gestational age (weeks) according to the last date of menstruation and the earliest ultrasound date available, and breastfeeding.

Infant intrauterine growth was classified according to international standards [27]. Those with a birth weight for sex and gestational age<10th percentile were defined as small for gestational age (SGA), between the 10th and 90th percentiles as adequate for gestational age (AGA), and >90th percentile as large for gestational age (LGA).

The BST provides a classification of performance in the motor (gross and fine), language (receptive and expressive), and cognitive subscales according to the cut-off point for age, defined as competent, emergent, and at-risk. We classified the infant at-risk if at least one at-risk test result was detected in any of the five domains, competent if no abnormality was detected in all domains, and emergent when a child was not at-risk and had one emergent result in at least one domain. 

Relative risks (RR) and 95% confidence intervals for emergent and at-risk categories were obtained by fitting simple and multiple log-multinomial regression models, considering alcohol and/or illicit drug(s) use during pregnancy, maternal schooling, prematurity, and intrauterine growth restriction as covariates. 

In the subgroup of 105 12-month-old ZIKV-exposed infants tested with the complete BSIDIII, we described the composite scores obtained in the cognitive, communication, and motor domains considering mild to moderate delayed infants with a raw score of 85–71 points, and severely delayed ≤ 70 points. We also described the scaled scores for five domains, considering mild to moderate delays as a score < −1 SD, and severe delays as a score < −2 SD.

## 3. Results

Maternal and infant characteristics are shown in Table 1. Maternal age and parity were slightly different between the groups. The frequency of alcohol and/or illicit drug use at least once a week, and gestational morbidities and use of alcohol and/or illicit drugs was higher in the ZIKV-unexposed group. ZIKV-exposed mothers had more than 12 years of schooling more frequently than ZIKV-unexposed mothers did. Most (84%) of the mothers acquired ZIKV infection in the second or third trimester of gestation. 

Neonates’ BW, sex, prematurity, intrauterine growth, and HC at birth were distributed similarly in both groups. Breastfeeding was more frequent in ZIKV-unexposed infants. None of the ZIKV-exposed infants failed to undergo hearing or eye screening. No severe hearing failure was detected, and few infants had minor ophthalmologic findings. 

Sixty infants from ZIKV-exposed cohort 1 received anti-ZIKV IgM testing within six months of age. Only two (3.3%) of them were positive. 

### 3.1. Risk of Neurodevelopment Abnormalities in ZIKV-Exposed and -Unexposed Children

The mean age at evaluation around 24 months of age was 23 and 22 months for ZIKV-exposed and -unexposed children, respectively. It is important to emphasize that the BST categorization into competent, emergent, or at-risk groups considers the children’s testing age. The HC measurements of children from both groups at the time of the BST were within the reference standards for age [28]. 

Table 2 shows classification data obtained from children according to their BST scores in different domains. In four of the five domains, few children were classified as emergent (4–12%) or at-risk (0.3–2.16%) independent of the maternal ZIKV exposure. However, higher proportions were found for the expressive communication domain: 19.1–42.9% emergent and 4.94–5.34% at-risk.

As shown in Table 3, ZIKV maternal infection was associated with children being half as frequently classified as emergent for neurodevelopmental delay, including after adjusting for covariates. However, no difference was detected in the at-risk category. Lower maternal education was associated with a higher likelihood of children being categorized as emergent but not at-risk.

Data obtained for the subset of 105 ZIKV-exposed infants submitted to the complete BSIDIII test (Appendix A) confirmed that 92–95% of these patients had adequate development up to 12 months of age. Mild to moderate delays were found in 3.81 to 9.80% of these infants in different domains, and 1% had severe delays in all domains. 

Notably, except for the expressive communication domain, the proportion of delays detected in this subset by the complete test is comparable to those caught as emergent or at-risk by the screening test around 24 months of age.

### 3.2. Risk of Being Classified as Emergent or At-Risk by BST in ZIKV-Exposed Children According to the Follow-Up Modality

The relative frequency of exposed children classified as emergent or at-risk in at least one of the five domains was similar in cohorts 1 and 2 of ZIKV-exposed children, even after adjusting for the covariates, showing that the outcome of children observed within six months of age (cohort 1) did not differ from that of children who started the follow-up later (cohort 2). In addition, the timing of ZIKV infection during pregnancy was not significantly associated with the outcome (Appendix A).

## 4. Discussion

We demonstrated that a large population-based cohort of non-microcephalic infants born to women with confirmed ZIKV infection during gestation were not at an increased risk of neuropsychomotor impairment up to 24 months of age compared to children born to mothers without ZIKV infection tested by the Bayley Scales of Infant and Toddler Development of Screening Test-3rd edition. ZIKV-exposed children were less likely to be classified as emergent in at least one of five domains than unexposed infants. These findings refute our a priori hypothesis and contrast with earlier reports that estimated a high risk of neurodevelopmental abnormalities in non-microcephalic ZIKV-exposed children at birth.

The original publications in this field described a cohort of ZIKV-exposed children from Rio de Janeiro, Brazil, born to symptomatic women with positive ZIKV nucleic acid testing. Using the complete Bayley-III scale, these studies have consistently reported below-average scores in at least one domain in approximately one-third of predominantly non-microcephalic children between 7 and 32 months [3,6,8,10] and at 3 years of age [9]. However, they lacked a control group, which precludes attributing neurodevelopmental delays to ZIKV exposure. Additional relevant factors for the observed delays include economic deprivation, which can profoundly affect neurodevelopmental outcomes, as demonstrated by the authors cited above [30] and others.

To date, few studies have compared the developmental outcomes of ZIKV-exposed normocephalic infants with those of their unexposed peers. Einspieler et al. compared the motor development of 76 12-month-old Brazilian infants with that of neurotypical Austrian infants, categorizing 18% as abnormal [13]. Poor performance in language, motor, and visual memory was found in 6-month-old Mexican infants born to symptomatic mothers than in those born to asymptomatic mothers [14]. Lower neurocognitive scores were detected in 32 exposed versus 97 unexposed, 24-month-old Nicaraguan children [15]. In contrast, Grant et al. did not observe differences in the proportion of delays between 154 exposed (15.3%) and 79 unexposed (25.3%) toddlers in French Guiana [18]. Likewise, 96 exposed Brazilian children had similar risks of neurodevelopmental delays as the 46 controls [19]. Using the full BSIDIII test, the authors also compared 235 exposed children with 39 controls and detected only a slight difference in cognition [16]. Analogously, 17 Brazilian 18-month-old infants performed similarly to 20 controls [20]. Blackmon et al. found that 68 of the exposed children had no developmental delays at two years of age, but they had more vision deficits than 63 of the unexposed children [17]. Limitations of the mentioned studies included the use of convenience samples with small sample sizes for cases and controls [14,15,20], or only for controls [16,19], defining the lack of maternal ZIKV infection using the absence of symptoms [14,20] or a single serological lab test at delivery [16,19], or ignoring potential confounders. 

We opted to use the BST in ZIKV-exposed children to take advantage of the availability of a control group from a large BRISA cohort. Although validation of this test and meaningful correlation with complete diagnostic tests have been demonstrated in Brazilian children [29,31], the BST provides a less detailed evaluation than the complete Bayley Scale, which is a diagnostic tool. However, the BST is suitable for identifying the risk of abnormalities using a controlled approach. Although using a different evaluation tool, our findings in exposed children being classified as emergent or at-risk in different development domains approached the reports from Brazilian (2.6–12.8%) [16], French Guinean (2.6–8.3%) [18], and Caribbean (0–19.0%) [17] cohorts of ZIKV-exposed children. In addition, our prevalence estimates of emergent or at-risk categorization in exposed infants (cognitive: 7.6%; fine motor: 5.6%, gross motor: 4,95; receptive communication: 12%, expressive communication: 24%) were within the boundaries of the pooled prevalence of any type of delay in the cognitive [6.5% (CI95% 4.1–9.3)], language [29.7% (CI95% 21.7–38.2)], and motor [11.5% (4.8–20.1)] domains reported by a meta-analysis of studies using BSIDIII scales in ZIKV-exposed children [21]. Of note, a high proportion of infants were categorized as emergent in the expressive communication domain, either in the exposed (19%) or in the unexposed (43%) infants. Although language is the domain in which most studies have reported delays, we suspect that it may have been difficult to apply or interpret this screening test modality, which depends on the infant’s expected language in specific situations. In addition, we did not confirm this finding in the subset of infants who underwent the complete Bayley test at 12 months of age, in which mild to severe delays in this domain reached 10%, but similar proportions of delayed children in other domains were found to be comparable to the BST. Furthermore, the distribution of delays in this subset of ZIKV-exposed children tested with the more in-depth BSIDIII is generally like that of other non-exposed Brazilian cohorts of typical children [32,33].

Importantly, our results indicate that asymptomatic infants whose mothers had Zika infection during pregnancy are not at an increased risk of delayed neurodevelopment up to 24 months of age, as suggested by others [16,17,18,19,20]. The at-risk classification proportions were similar between the groups. However, ZIKV exposure functioned as an independent protective factor against emergent classification. The reasons for this finding remain unclear. We speculate that public information and health services advice on the potential adverse consequences of neurodevelopment resulted in increased parental stimulation of the infant during the last Zika virus epidemic. During the follow-up, we witnessed deep family concerns about their ZIKV-exposed child’s development, even if they were not apparently affected, while this was not noted for the ZIKV-unexposed cohort. This fact is indirectly reinforced by our findings of the similar performance of the ZIKV-exposed children, independent of follow-up regularities with developmental specialists. Alternatively, the non-contemporary evaluation of cohorts might have resulted in measurement artifacts across the groups. Although the staff executing the BST differed between the BRISA and NATZIG cohorts, they were trained and supervised for quality assurance. Other factors, such as family socioeconomic status and home environment, which were not assessed in this study, could have affected the outcomes. Additional studies will help us to better understand our findings. 

The main strength of our study is the robust sample size of both groups of children. The study group constituted the largest population-based cohort of infants born to mothers with symptomatic confirmed ZIKV infection, evaluated for neurodevelopment at 24 months. All of the children lived in the same population with similar sociodemographic backgrounds. We could define a large control group because no ZIKV was circulating in Brazil. Therefore, there was no need to rely on cross-reactive serological testing. Both cohorts had information on the relevant cofactors for child outcomes. Additionally, we did not include infants with other health conditions that could affect the outcome evaluation. 

Study limitations include the historical non-concurrent control group and enrollment of only symptomatic women who had confirmed infection by molecular testing, an underrepresentation of all women with ZIKV infection in pregnancy. Selection bias could have occurred in cases where maternal asymptomatic infection differed in causing CNS abnormalities in infants. However, because we could enroll all those with a confirmed infection, we believe our sample is representative of symptomatic women from this population. The influence of repeated testing and maternal instruction on stimulating their child’s neurodevelopment in the NATZIG cohort 1 should be addressed. Training occurred, and improved child and maternal comfort with the test and examiner may have influenced the results. Yet, we did not find differences in the proportion of children classified as emergent/at-risk in cohorts 1 and 2. These factors and selecting a historical control as the unexposed group might have led to underestimating the exposure–outcome relationship. Although we used an objective method to define the lack of microcephaly in newborns, the absence of CNS lesions was confirmed in only half of the exposed babies by cranial ultrasound, and we did not perform imaging examinations in the control group. Our results are generalizable to normocephalic infants without apparent abnormalities at birth, born to women with symptomatic ZIKV infection who do not develop microcephaly within 24 months of age. 

In conclusion, our findings show that, compared to unexposed children, ZIKV-exposed children are not at an increased risk of neuropsychomotor impairment up to 24 months of age. 

## Figures and Tables

**Figure 1 pathogens-12-01219-f001:**
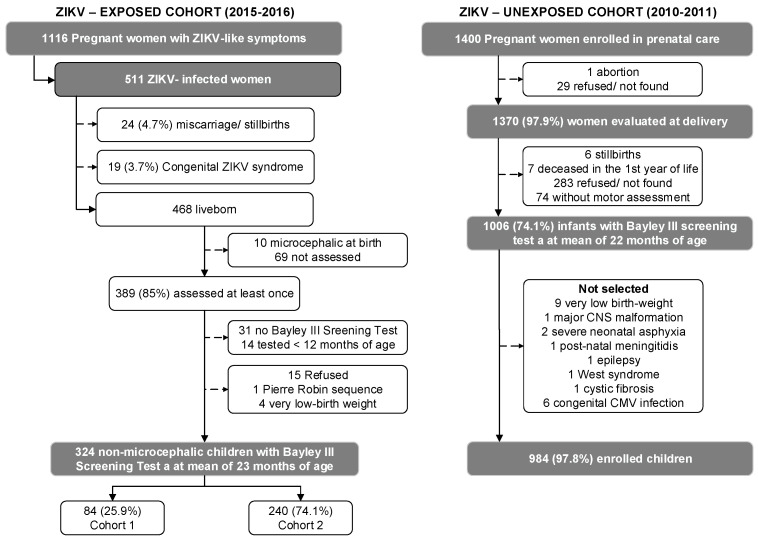
Derivation of the ZIKV-exposed and ZIKV-unexposed cohorts of infants evaluated around 24 months of age. Cohort 1 of exposed infants were those followed periodically from birth. Cohort 2 of exposed infants were those not observed early on and evaluated at least once at 24 months of age.

**Table 1 pathogens-12-01219-t001:** Maternal and infants’ characteristics according to exposure to ZIKV during pregnancy.

	ZIKV-Exposed (NATZIG Cohort, n = 324)			ZIKV-Unexposed (BRISA Cohort, n = 984)
Mothers	Cohort 1 (n = 84)	Cohort 2 (n = 240)	Total (n = 324)	Total (n = 984)
Age (years)	26.76 [6.45]	28.00 [5.95]	27.67 [6.10]	26.10 [6.13]
Years of schooling				
>=12	16 (19.05)	71 (30.47)	87 (27.44)	79 (8.04)
9 to 11	45 (53.57)	99 (42.49)	144 (45.43)	638 (64.90)
<=8	23 (27.38)	63 (27.04)	86 (27.13)	266 (27.06)
Primiparous	35 (44.30)	80 (35.24)	115 (37.58)	438 (44.51)
Hypertension and/or Mellitus diabetes #	13 (15.48)	38 (16.17)	51 (15.99)	185 (19.03)
Alcohol/illicit drugs use at least once a week	20 (23.81)	54 (22.98)	74 (23.27)	303 (30.82)
Trimester of ZIKV infection				
First	7 (8.75)	29 (12.39)	36 (11.50)	-
Second	35(43.75)	100 (42.74)	135 (42.99)	-
Third	38 (47.50)	105 (44.87)	143 (45.69)	-
**Children**				
Birth weight (g)	3263.7 [419.73]	3199.17 [434.88]	3215.76 [431.3]	3235.93 [472.58]
Male sex	47 (55.95)	131 (55.27)	178 (55.45)	480 (48.78)
Prematurity (<37 weeks)	1 (1.19)	22 (9.21)	23 (7.12)	55 (5.59)
Intrauterine growth restriction	8 (10.13)	16 (6.87)	24 (7.69)	79 (8.06)
Head circumference at birth (cm)	34.51 [1.09]	34.29 [1.34]	34.35 [1.28]	34.30 [1.63]
Age at BST (months)	23.21 [2.04]	23.48 [1.79]	23.41 [1.85]	22.41 [3.29]
Head circumference at the BST (cm)	47.88 [1.26]	47.92 [1.79]	47.91 [1.67]	48.25 [1.78]
Exclusive or mixed breastfeeding by 6 months	-	-	150 (44.37)	699 (70.60)
Cranial sonography				
Normal	28 (33.33)	62 (25.83)	90 (27.78)	-
Minor abnormalities ^1^	19 (22.62)	39 (16.25)	58 (17.90)	-
Not done	37 (44.05)	139 (57.88)	176 (54.32)	-

Frequency (%), Mean [SD], ^1^ periventricular/subependymal cyst, pericerebral minor enlargement, and/or lenticulostriate vasculopathy, BST = Bayley III Screening Test, # previously or during gestation.

**Table 2 pathogens-12-01219-t002:** Absolute and relative frequencies for children classification on each of the domain’s subscales of the BSIDIII Screening test, according to maternal ZIKV infection exposure.

Domain Categories	TOTAL	ZIKV-Exposed	ZIKV-Unexposed
	N = 1308	N = 324	N = 984
** *Cognitive* **			
Competent	1147 (87.83)	299 (92.28)	848 (86.35)
Emergent	143 (10.94)	22 (6.79)	121 (12.32)
Risk	16 (1.23)	3 (0.93)	13 (1.32)
Missing	2	0	2
** *Fine motor* **			
Competent	1208 (92.49)	307 (94.75)	901 (91.75)
Emergent	92 (7.05)	16 (4.94)	76 (7.74)
At-Risk	6 (0.46)	1 (0.31)	5 (0.51)
Missing	2	0	2
** *Gross motor* **			
Competent	1205 (92.27)	307 (95.05)	898 (91.35)
Emergent	91 (6.97)	14 (4.33)	77 (7.83)
At-Risk	10 (0.76)	2 (0.62)	8 (0.81)
Missing	2	1	1
** *Receptive Communication* **			
Competent	1181 (90.36)	285 (87.96)	896 (91.15)
Emergent	111 (8.49)	32 (9.88)	79 (8.04)
At-Risk	15 (1.15)	7 (2.16)	8 (0.81)
Missing	1	0	1
** *Expressive communication* **			
Competent	750 (57.78)	246 (75.93)	504 (51.75)
Emergent	480 (36.98)	62 (19.14)	418 (42.92)
At-Risk	68 (5.24)	16 (4.94)	52 (5.34)
Missing	10	0	10

**Table 3 pathogens-12-01219-t003:** Absolute and relative frequencies and relative risks for classification of children on the BSIDIII Screening test, according to the participants characteristics.

	Competent (N = 703) n (%)	Emergent (N = 515) n (%)	At-Risk (N = 79) n (%)	Crude RR (95% CI)/ Adjusted RR (95% CI) for Emergent	Crude RR (95% CI)/ Adjusted RR (95% CI) for At-Risk
Maternal ZIKV infection ^Ω^					
Yes	235 (72.53)	72 (22.22)	17 (5.25)	**0.48 (0.39; 0.61)/** **0.52 (0.40; 0.66)**	0.82 (0.49; 1.39)/ 0.83 (0.48; 1.44)
No	468 (48.10)	443 (45.53)	62 (6.37)	ref	ref
Maternal schooling (years) ^£^					
<=8	183 (52.59)	139 (39.94)	26 (7.47)	**1.64 (1.22; 2.22)/***	1.23 (0.61; 2.50)/*
9 to 11	398 (51.29)	335 (43.17)	43 (5.54)	**1.78 (1.34; 2.36)/***	0.91 (0.47; 1.78)/*
>=12	115 (69.70)	40 (24.24)	10 (6.06)	ref	ref
Maternal Alcohol/illicit drugs ^#¥^					
No	506 (55.18)	353 (38.50)	58 (6.32)	ref	ref
Yes	192 (51.34)	161 (43.05)	21 (5.61)	1.12 (0.97; 1.29)/*	0.89 (0.55; 1.44)/*
Prematurity (<37 weeks) ^∞^					
No	660 (54.19)	483 (39.66)	75 (6.16)	ref	ref
Yes	42 (53.85)	32 (41.03)	4 (5.13)	1.03 (0.78; 1.36)/*	0.83 (0.31; 2.22)/*
Intrauterine Growth restriction ^€^					
No	641 (54.28)	469 (39.71)	71 (6.01)	ref	ref
Yes	49 (49.00)	44 (44.00)	7 (7.00)	1.11 (0.87; 1.40)/*	1.16 (0.55; 2.46)/*

ref = reference. Crude RR = relative risk for each category of the BSIDIII Screening test results, showing the reference group. Adjusted RR = RR fitting multiple log multinomial regression models, considering alcohol and/or illicit drug(s) use during pregnancy, maternal schooling, prematurity, and intrauterine growth restriction as covariates; 95% CI = 95% confidence interval, ^#^ at least once a week, * not done. Missing values absolute frequency: ^Ω^ (11); ^£^ (8); ^¥^ (6); ^∞^ (1); ^€^ (16). Results shown in bold are those with the 95% confidence intervals that do not contain the number one.

## Data Availability

The data that support the findings of this study are available from the corresponding author upon reasonable request.

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
