# Peer review of "Normocephalic Children Exposed to Maternal Zika Virus Infection Do Not Have a Higher Risk of Neurodevelopmental Abnormalities around 24 Months of Age than Unexposed Children: A Controlled Study"

_pathogens, 2023, doi:10.3390/pathogens12101219_

Round 1

Reviewer 1 Report

This paper reports on a study comparing neurodevelopmental outcomes in children exposed prenatally to Zika virus versus unexposed children in Brazil. 324 infants born to mothers with confirmed Zika infection were compared to 984 infants born prior to the Zika outbreak on performance on a standardized screening test of development at around 24 months of age. Contrary to expectations, the study found that the exposed children were significantly less likely to be classified as having delays in development compared to unexposed children. The authors suggest this may be due to increased stimulation and intervention for the Zika-exposed children. They conclude that non-microcephalic infants prenatally exposed to Zika do not appear to be at higher risk for neurodevelopmental abnormalities in the first two years of life compared to unexposed peers from the same population. It would be an addition to the paper if known to provide the gestational stage at which the mother was infected with ZikaV. It is hard to tell if these were infections at the time of birth which would potentially limit the effects of virus on the neonatal brain and be relevant to the drawn conclusions.

some items to consider for possible expansion or future studies,

  • The control group is not contemporaneous, being enrolled 6 years earlier. Changes over time could account for some differences between groups.
  • Exposed mothers were symptomatic and tested positive for Zika. The results may not generalize to infants born to women with asymptomatic infection.
  • Socioeconomic status and home environment were unaccounted for, both of which impact development. The control group appears to have had higher alcohol/drug use rates, suggesting possible socioeconomic differences.
  • The screening test used provides less in-depth assessment than a full diagnostic battery. Subtler delays may have been missed.
  • No imaging was done in controls to definitively rule out subclinical brain abnormalities.

The finding that exposed infants performed better than expected is surprising. The authors suggest increased stimulation efforts for exposed children could account for this. However, this is speculative.

Overall, the study benefits from a strong sample size and control group. However, differences between groups beyond Zika exposure and limitations in developmental screening measures make conclusions about Zika's neurodevelopmental effects tentative. Additional research controlling for confounds and using full diagnostic assessments could build on these results. The implications for long-term outcomes as children mature also remain to be investigated.

Author Response

1- It would be an addition to the paper if known to provide the gestational stage at which the mother was infected with ZikaV. It is hard to tell if these were infections at the time of birth which would potentially limit the effects of virus on the neonatal brain and be relevant to the drawn conclusions.

Response - As shown in Table 1, maternal ZIKV infections occurred in the first (11.50%), second (42.99%), or third trimesters of gestation (45.69%). No mother had the infection at the time of birth. We tested if the timing of maternal ZIKV infection (Table S2) was associated with the outcomes of children being classified as emergent or at risk by the Bayley Screening Test and did not find such an association. 

2- Some items to consider for possible expansion or future studies

Response- Unfortunately, we will not be able to expand the study to exclude the limitations, we did our best with the available data.

3- The control group is not contemporaneous, being enrolled 6 years earlier. Changes over time could account for some differences between groups.

Response: We recognized this limitation in the Discussion, lines 316, 326.

4- Exposed mothers were symptomatic and tested positive for Zika. The results may not generalize to infants born to women with asymptomatic infection

Response: This limitation is emphasized in the Discussion on lines 316-319

5- Socioeconomic status and home environment were unaccounted for, both of which impact development.

Response - We agree with the reviewer. This was recognized in the Discussion on lines 302-304. We used the maternal years of schooling to indirect represent this variable. Although not perfect, it is frequently use as a proxy for socioeconomic status. We detected that lower maternal education was associated with a higher likelihood of children being categorized as emergent but not at-risk.

6- The control group appears to have had higher alcohol/drug use rates, suggesting possible socioeconomic differences

Response - This is correct. However, we did not find an association of this variable with the BST results. Also, the multinomial regression model to define the effect of the maternal ZIKV infection on the children`s BST classification considered  alcohol and/or illicit drug(s) use during pregnancy as well as other variables (maternal schooling, prematurity, and intrauterine growth restriction) as covariates, to control for the imbalance or possible biological effects on the outcome. 

7- The screening test used provides less in-depth assessment than a full diagnostic battery. Subtler delays may have been missed.

Response: We agree with the reviewer. As the manuscript (lines 264-267) states, the BST provides a less detailed evaluation than the complete Bayley Scale, a diagnostic tool. However, the BST is suitable for identifying the risk of abnormalities in a controlled approach. Our objective was to compare the performance of ZIKV-exposed and ZIKV-unexposed infants in BST to use it as a marker for the risk of neurodevelopment delay. The robust sample size gave us estimations with small Confidence intervals.

8-No imaging was done in controls to definitively rule out subclinical brain abnormalities.

Response: This is true. However, considering that the frequency of brain major abnormlities in the general population is low and the big sample size of the studied group, it is likely that  measurement bias would be minimal. We added a phrase to the Discussion (line 332-333)

9-The finding that exposed infants performed better than expected is surprising. The authors suggest increased stimulation efforts for exposed children could account for this. However, this is speculative.

Response: We agree with the reviewer. We used the word speculate on line 293 when we give this possible explanation. 

10- The finding that exposed infants performed better than expected is surprising. The authors suggest increased stimulation efforts for exposed children could account for this. However, this is speculative.

Overall, the study benefits from a strong sample size and control group. However, differences between groups beyond Zika exposure and limitations in developmental screening measures make conclusions about Zika's neurodevelopmental effects tentative. Additional research controlling for confounds and using full diagnostic assessments could build on these results. The implications for long-term outcomes as children mature also remain to be investigated.

Response: - We agree with the reviewer. However, we will need to wait for another ZIKV epidemic wave to be able to do so. Nevertheless, our data was obtained from well-controlled cohorts with robust sample sizes carefully analyzed to adjust for available confounders. It will be helpful on clinical and population grounds. 

Thank you

Reviewer 2 Report

In this manuscript, the authors present the results of the analysis of two integrated cohorts of normocephalic children born to ZIKV-exposed pregnant women as compared to a historical cohort of children born to ZIKV-unexposed pregnant women (recruited and followed before the epidemic of Zika) in terms of their neurodevelopment as measured by the Bayley Screening Test III around 24 months of age. The study was based on an appropriately formulated research question, aimed at a well-known knowledge gap, and provided high-quality evidence.

Methods

The methodology followed by the authors was clearly and comprehensively declared. It was also appropriate to deal with the data and problem: exposure and outcome were determined using standardized and validated methods. There are two concerns related to the potential introduction of bias: first, the exclusive inclusion of symptomatic pregnant women might have introduced selection bias as asymptomatic cases of gestational infection might be at lower risk of developing children’s nervous system abnormalities. In this case, an overestimation of the relationship between exposure and outcome should be expected. Second, the authors state, "At each visit, the study staff instructed the mothers on how to stimulate their child’s neurodevelopment using national guidance.”. That is, about 1 out of 4 exposed mothers (cohort 1) were given additional information about how to stimulate their children during the follow-up, and more generally (see discussion), exposed pregnant women during the epidemic were aware of the potential effect(s) of the infection on their children’s neurodevelopment. This, and selecting a historical cohort as the unexposed group, might have led to underestimating the exposure-outcome relationship. Notably, the authors demonstrated that both exposed cohorts had similar characteristics regardless of the differences in the follow-up protocols. Finally, regarding confounding, the authors did their best to control for covariates with known effects on the outcome that were differentially distributed between the study groups.

Results

The results of the study were clearly and comprehensively presented.

Discussion/conclusion

The authors summarized the study's main findings, critically appraising and carefully framing them in the context of the available evidence. The strengths and limitations of the design were thoroughly discussed, and the conclusion was consistent with the results.

Author Response

With respect to the concerns related to the potential introduction of bias

1- First, the exclusive inclusion of symptomatic pregnant women might have introduced selection bias as asymptomatic cases of gestational infection might be at lower risk of developing children’s nervous system abnormalities. In this case, an overestimation of the relationship between exposure and outcome should be expected.

Response- We agree with the reviewer. In the Discussion session we recognize that limitation: Study limitations include the historical non-concurrent control group and enrollment of only symptomatic women who had confirmed infection by molecular testing an underrepresentation of all women with ZIKV infection in pregnancy. Selection bias could have occurred  in case maternal asymptomatic infection differed in causing CNS abnormalities in infants. (page 9, lines 313-316)

2-Second, the authors state, "At each visit, the study staff instructed the mothers on how to stimulate their child’s neurodevelopment using national guidance”. That is, about 1 out of 4 exposed mothers (cohort 1) were given additional information about how to stimulate their children during the follow-up, and more generally (see discussion), exposed pregnant women during the epidemic were aware of the potential effect(s) of the infection on their children’s neurodevelopment. This, and selecting a historical cohort as the unexposed group, might have led to underestimating the exposure-outcome relationship.

Response- Thank you for bringing up this aspect. We additionally commented this limitation in the Discussion Section. It now reads:  "The influence of repeated testing and maternal instructing on how to stimulate their child’s neurodevelopment in the NATZIG cohort 1 should be addressed. Training occurs, and improved child and maternal comfort with the test and examiner may influence results. Yet, we did not show differences in the proportion of children classified as emergent/at-risk in Cohorts 1 and 2; these facts and selecting a historical control as the unexposed group might have led to underestimating the exposure-outcome relationship." (lines 317-326).